# Oral Manifestations of Rett Syndrome—A Systematic Review

**DOI:** 10.3390/ijerph18031162

**Published:** 2021-01-28

**Authors:** Syed Sarosh Mahdi, Hafsa Abrar Jafri, Raheel Allana, Francesco Amenta, Mariam Khawaja, Syed Saad B. Qasim

**Affiliations:** 1Department of Community Dentistry, Jinnah Medical and Dental College, Sohail University, Karachi 74800, Pakistan; hafsajafri1@gmail.com (H.A.J.); mariamkhawaja12345@gmail.com (M.K.); 2Athena Center for Advanced Research in Healthcare, 62032 Camerino, Italy; 3Centre of Clinical Research, Telemedicine and Telepharmacy, School of Medicinal and Health Products Sciences, University of Camerino, 62032 Camerino, Italy; famenta@gmail.com; 4Department of Paediatrics & Child Health, Aga Khan University Hospital, Karachi 74800, Pakistan; dr.raheelallana@hotmail.com; 5Department of Bio Clinical Sciences, Faculty of Dentistry, Kuwait University, Kuwait City 12037, Kuwait; sayed.qasim@hsc.edu.kw

**Keywords:** bruxism, rare disorders, dental education

## Abstract

Rett Syndrome is an x linked developmental disorder which becomes apparent in females after 6 to 18 months of age. It leads to severe impairments including loss of speech, loss of hand movements/manual dexterity, characteristic hand movements such as hang wringing and intellectual disability/learning problems. This systematic review was carried out to identify the dental manifestation of Rett syndrome and to shed light on treatment options available for oral health problems associated with Rett syndrome. A systematic literature search was conducted on the PubMed, Scopus, Biomed, Web of Science, Embase, Google Scholars, Cochrane and CINAHL using the following entries: Rett syndrome (*n* = 3790), Oral health and Rett syndrome (*n* = 17), dental health of Rett syndrome patients (*n* = 13), and the MeSH terms listed below: Rett syndrome and Oral Health (*n* = 17), Rett syndrome and dentistry (*n* = 29). The final review included 22 search articles. The most common oral findings was bruxism. Masseteric hypertrophy was also reported. Anterior open bite and non-physiological tooth wear was observed. Other oral manifestations of Rett syndrome included mouth breathing, tongue thrusting, digit/thumb sucking, high arch palate. Increased awareness and dental education amongst dentists and assistants regarding the dental manifestations of Rett syndrome and similar neurodevelopmental disorders is required to improve the level of care and empathy they can provide to these differently able patients. Research on dental aspects of Rett is scarce and this remains a neglected topic.

## 1. Introduction

Neurodevelopment has been referred to as an intricate and dynamic correlation in between genes, brain, cognitive, emotional and behavioral process across the developmental lifespan. A significant and repetitive disruption to this dynamic interplay through environmental and genetic risk can eventually lead to neurodevelopmental disabilities. Such disorders have been linked with low income communities and children living in poverty [1]. Conditions such as autistic spectrum disorder, Aspergers syndrome, attention deficit hyperactivity disorder, dyspraxia and Rett Syndrome are characterized as being neurodevelopment disorders [2]. Among these, Rett Syndrome has been exclusively linked with female predilection and mental retardation [3].

It was first identified by an Austrian pediatric neurologist called Dr. Andres Rett in 1966. However, the first published report was noted in 1983 by Dr. Bengt Hagberg et al., [4] who reported a pooled investigation of 35 patients that lead to a significant increment in interest and awareness about this rare condition [5]. A recent report by Alan Percy mentioned that there has been a remarkable progress made over the past few decades with respect to Rett Syndrome [4], specifically after the discovery of its correlation with mutations in the X-linked gene encoding methyl-CpG-binding protein 2 gene (MECP2). This disease is characterized to occur in different stages. An initial stage of early normal development is followed by regression phase appearing as severe intellectual disability. Hagberg outlined four classical stages of Rett syndrome that started from a delayed developmental milestone to further regression with loss of acquired communication and signs of mental retardation. These were followed by difficulties in walking and neuromotor retardation followed by a later stage of motor deterioration characterized by weight loss and skeletal deformities [6].

Few other cardinal features also include characteristic teeth grinding, altered breathing patterns like hyperventilation and breath holding, poor sleep patterns, feeding and gastrointestinal problems, occurrence of seizures, scoliosis [7], small feet and hands, failure to thrive and the development of osteoporosis. To date there is no cure reported for Rett Syndrome. However a palliative treatment strategy can alleviate functional, sensory and motor skills. This has led to a survival rate of at least six decades [8].

With respect to the oral health conditions of patients with Rett Syndrome, limited reports have been able to describe dental characteristics. Bruxism has been commonly reported as a repetitive feature, however this is not pathognomonic [6,9]. Other dental findings reported are gingivitis, poor periodontal status, anterior open-bite is also noted to be associated with palatal shelving. This could be due to mouth breathing, digital sucking and mouthing. Other dental anomalies include malocclusion resulting from incorrect mandibular positioning and unfavorable growth patterns [6].

The aim of the current systematic review is to deliver an evidence-based update on the oral and dental manifestation associated with Rett Syndrome and to improve awareness on oral aspects of this rare disease.

### Research Question

A research question was created based on the Preferred Reporting Items for Systematic Review and Meta-Analysis (PRISMA) guidelines [10]. The focused question formulated was to ask what the oral health manifestations associated with Rett Syndrome are.

## 2. Methodology

General search of the data was conducted, various databases including PubMed, Scopus, Cochrane, Google scholar, CINAHL, Embase, and Web of Science and Biomed databases were searched related to this topic.

### 2.1. Search Strategy and Study Selection

A multicomponent search strategy was employed using multiple databases to increase the sensitivity and reach of the review. Search strategies used in previous systematic reviews were examined and updated to include studies [11] relevant to this review. General search terms and MeSH (Medical subject heading) were used. The following entries were used for searching, Rett syndrome (*n* = 3790), dental health of Rett syndrome patients (*n* = 13), Rett syndrome and Oral Health (*n* = 17), Rett syndrome and dentistry (*n* = 29) (Prisma flowchart Figure 1, (Table 1). A MeSH (Medical Subjects heading) search strategy did not return any article. The search findings were downloaded into EndNote bibliographic software and duplicates were eliminated. The reference lists of studies that filled the eligibility criteria were also searched. We organized the search and selection of studies following the SPIDER question format adopted from the PICO tool (Figure 2). The search framework we employed for our inclusion criteria were that all articles selected should be in the English language and that keywords should match keywords in the title or the abstract of the studies selected for our review (Table 2). The following types of studies were included in our review based on our keywords and criteria: “original articles, systematic reviews/meta- analysis, case reports and case series”. Pilot studies, correspondences and editorials were not included. The scarcity of articles on dental aspects of Rett syndrome implied that diverse types of studies had to be included in the review following compliance with Quality assessment protocols. Four reviewers (Drs. SM, HJ/RA and FA) examined the titles and abstracts of all relevant articles to ascertain the criteria of our exclusion and inclusion benchmarks (Table 2). Articles published before 1990 were not included. Only articles published in peer-reviewed journals were selected for the final review.

All difference of opinions regarding the selection of studies were resolved via discussion. The parameters the researchers applied were: English language, at least one keyword corresponding to the above entries in the title/abstract and study based on the evaluation of clinical trials. Moreover, books, and other online materials of interest on the topic were selected and examined. A further analysis was performed by a researcher (Dr. RA) using a conventional approach consisting of reading the title and the abstract of the paper and only selecting less than the 10% of papers identified. An additional evaluation criterion was the publication of papers in peer-reviewed journals. The literature focusing on dental aspects of Rett Syndrome was extremely sparse and most of the articles related to non- dental manifestations of Rett Syndrome. The Prisma 2009 (preferred reporting items for systematic reviews and meta-analysis was also complied with through the process of study selection and search strategy (Prisma flowchart Figure 1). With respect to the Exclusion criteria, studies or other materials published before 1990 were not included in our analysis. The reason for excluding older studies was primarily that not much literature exists before the 1990s and including outdated data due to the advancement in medicine since the 1990′s makes it a counter-productive exercise. Therefore, the selected articles were published between the years 1990 and 2019. Papers published in non-peer-reviewed journals were also excluded and pilot experiments were discarded due to the limited number of patients examined in these pilot studies. Non-English studies were negligible in number and a systematic review has concluded that including language restriction in a systematic review does not lead to systematic bias [12]. Finally, the articles without the keywords Rett syndrome/oral hygiene, oral health/dentistry within the title and/or in keywords were discarded (Table 2).

### 2.2. Quality Assessment

The relevant articles extracted were evaluated with the Newcastle–Ottawa scale of assessing the quality non-randomized studies [13]. A ten point/star* modified version of the Newcastle–Ottawa scale was utilized to assess the quality of the studies included in the review. The scoring was guided by the criteria laid out by the scale. The study quality was designated as poor (0–4), moderate (5–6) and good (7–9) (Table 3). The score was based on attributes that could be credited to outcomes (i.e., selection, comparability, outcome with the related features). Additionally, another tool, the Oxford levels of evidence criteria, was used to assess the quality of the clinical evidence presented in the evaluated studies for all types of studies. Overall the study levels of quality was dependent on the type of study conducted, whether it was an original study (RCT, case control/cohort, cross sectional), a meta-analysis, case series or case reports (Table 4).

## 3. Results

Out of 22 search articles reviewed, the most common findings were bruxism, (Peak et al. [14], Ribeiro et al., 1997 [15], Alpoz et al., 1999 [16], Magalhães 2002 [18], Khalil, 2002 [19], Manish, 2010 [24], Ji Sung Nho et al., [25], Fuertes et al., 2014 [28], Janas, 2015 [30], Lai YYL, 2018 [34]). Massetric hypertrophy was also reported by Peak et al. [14]. Anterior open bite was reported Ribeiro et al., 1997 [15] and Fuertes et al., 2014 [28]. Non-physiological tooth wear was observed by Peak et al., 1992 [14], Ribeiro et al., 1997 [15], Fuertes et al., 2014 [28] (Table 5). Other findings along with bruxism and tooth wear included mouth breathing, tongue thrusting, digital sucking, high arch palate (Riberiro et al., 1997 [15] and Fuertes et al., 2014 [28]. Trauma to the oral soft tissues was also evident with tissue changes including gingivitis and hypertrophic gingival tissues [15,28,30]. Dental caries was evident in most patients [34], (Table 5). Fourteen studies were assessed to be of Good Quality (7–9*) according to the Newcastle–Ottawa scale [14,15,19,20,21,22,25,26,28,29,32,33,34,35], seven studies were found to be of moderate quality (5–6*), [16,17,18,23,24,30]. One study was deemed to be of poor quality according to the scale [31].

## 4. Discussion

### 4.1. Oral Clinical Manifestations of Rett Syndrome

Bruxism was found to be the most common oral finding in almost all the Rett syndrome patients. Intriguingly, nine studies identified bruxism as the most prevalent oral manifestation of RTT [14,15,16,18,19,25,28,29,34]. Amongst the type of bruxism, diurnal bruxism was more prevalent than the nocturnal one. Temudo et al. stated that diurnal or day time bruxism seemed to suggest the existence of an MECP2 mutation in a child with RTT [36]. Out of 13 patients, Magalhaes et al. [18] reported 11 patients with daytime bruxism as the primary oral trait. Fuertes et al. [28] investigated a group of patients with Rett syndrome and one control group comprising healthy patients, and a prospective case control study was carried out by means of a questionnaire and detailed dental examination. The most relevant findings in the Rett syndrome case group were diurnal bruxism, which comprised of 68.3% and the nocturnal type was prevalent in 4.9%. The process by which the patient grinds her teeth tends to indicate that the forward orientation of the teeth during the movement of the tooth grinding entitles the patient towards anterior tongue thrusting [15].

Other studies shed light on the fluency and the communication problem which the Rett patients experienced in their daily life. A questionnaire-based study was performed by Lavas et al. [21], which involved caregivers of 125 rett patients. This study described communication problems as the greatest problem faced by caregivers, their verbal vocabulary was restricted. For effective intervention, it is recommended that family and clinicians engage closely to determine the communication abilities of patients with Rett syndrome. Manish et al. [24] presented a case of an eight-year-old female child who was presented in the department of psychiatry at Bathla Hospital. She had a loss of speech and her social interaction was also limited. Da Silva et al. reported the integration between speech and dentistry. Furthermore, they mention about the stomatognathic system in which both the speech pathologist and the dentist complement each other to achieve better clinical outcomes [37].

Some studies also reported non-physiological tooth wear among Rett patients [14,28]. Rebeiro et al. [15] identified attrition in 71% of the children. Acid reflux has been generally described in Rett patients [38] and may lead to dental erosion, but to date, no studies have directly linked non-physiological tooth wear to erosion due to gastroesophageal reflux. However, evidence from the literature suggests that there is a bidirectional relationship between gastro-intestinal reflux and bruxism [39]. A recent systematic review reported that there is a 2 to 4 fold increment in the odds ratio for individuals presenting with erosive tooth wear in relation to GERD. Moreover, individuals with erosive tooth wear will have a negative impact on the quality of life. Hence, referrals to oral health care professionals should be considered from a gastroenterology point of view and vice versa [40]. Other assorted oral findings found consistently in the literature include tooth wear, mouth breathing, tongue thrusting, digital sucking and high arch palate [15,28]. Fuertes et al. [28] identified tongue thrusting in 56.1% of Rett patients. These parafunctional habits trigger or worsen the anterior open bite as well as palatal shelving.

Among the soft tissue changes, Gingivitis and periodontal changes were the most common [15,28]. These findings are consistent with the knowledge available in the literature. In effect, periodontal disease findings of mentally impaired patients (mental retardation not associated with Down’s syndrome and autism) regularly report elevated incidence and severity of gingivitis and periodontitis secondary to dental plaque deposition relative to the general population [41]. As a result, the caregiver can have trouble preserving an adequate standard of oral hygiene, considering the difficulties of accessing an individual’s oral cavity, which can be further exacerbated by their behavioral characteristics [15].

Y.Y.L Lai et al. [34] described the incidence of caries and restorations in a 242 female cohort. The incidence rate for restored teeth was 6.8 per 100 person per year and rate of extractions was 9.3 per 100. This was the first study which studied restorations and dental caries rates in a population. Just a limited number of carious lesions have been identified in patients with Rett syndrome [15,28] and some Rett syndrome patients have been free of caries [14]. It has been proposed that the low incidence of active caries is attributed to the more intensive treatment of patients with Rett syndrome under general anesthesia in order to prevent repeat recurrence of anesthetics and treatment [21].

However, all these studies failed to discuss whether all these oral findings occurred due to Rett syndrome or the sugar content the medications these patients were receiving for their illness or due to their oral hygiene practices.

### 4.2. Management of Oral Manifestations in Rett Syndrome Patients

There are two types of oral manifestations which are linked with Rett syndrome, one which is due to the drugs, e.g., anticonvulsant drugs, which includes dryness of the mouth (xerostomia), glossitis, gingival hyperplasia, parotitis, sialorrhea, etc.

As far as dental aspect is concerned, bruxism, gingivitis and dental cavities are most prevalent among the other dental conditions that occur with Rett syndrome, these conditions should be carefully examined and promptly treated during the course of this syndrome and timely interventions should be carried out to prevent these conditions. Bruxism should be reduced in order to prevent further wearing off the teeth surfaces, however Fuertes et al. have contraindicated the use of splints due to their incontrollable motor function in these cases [26]. Alpoz et al. [16] used splints which were soft in consistency to treat bruxism, however specialized designs such as bite planes were constructed. Modified splints were used in the upper jaw to ensure that the cuspids of the deciduous lower molar teeth displayed centric occlusion in relation to the splint. Denture adhesives were used to increased retention and the margins of the splint were extended till the hard palate. The occlusal surface of the splint was modified with a groove, which measured 5 mm in width and was 1.5 mm deep. Resilient material was used to fill the groove to prevent anterior mandibular movement. Acupuncture was used in combination with this treatment and was reported to be effective against temporomandibular problems and facial pain. The caregivers reported positive results from these two combined treatment regiments and parents reported that bruxism did not appear seven days following acupuncture sessions and even when it returned the intensity was reduced. The combination of modified and acupuncture for the management of bruxism was considered effective by the authors [16].

Dental caries in Rett syndrome patients is very common and it requires the cooperation of parents of these children. Fluoride applications should be used to reduce the likelihood of caries and frequent visits should also be encouraged to dentists for follow up [41].

### 4.3. Behavioral and Pharmacological Management of Rett Syndrome Patients

Various therapeutic and pharmacological treatment methods have been reported in the literature for Rett patients. The use of mouth props during oral examination or treatment was in general [42]. Several scholars have proposed that dental management should be carried out under sedation. Peak et al. [14] advocated the use of analgesia, sedation and even general anesthesia when appropriate, due to the difficulties of delivering dental care under local anesthesia. Ji Sung Nho et al. [25] presented a case of a 19-year-old girl. She was scheduled for an elective procedure for intraocular lens insertion under General Anesthesia. The author/clinician faced difficulties in the airway management of this patient due to breathing abnormalities and the presence of muscular tonicity. He suggested that anesthetists should have proper knowledge in order to avoid complexities in Rett syndrome patients during their management. Omer et al. [32] presented a case of a girl who was 4.5 years old. She lacked cooperation for any dental restoration under local anesthesia so she was scheduled for sedative analgesia. The author favors local anesthesia whenever possible in patients with Rett syndrome, since it reduces demand for opioids, anesthetics and postoperative care, but anesthesiologists should have full knowledge regarding the Rett syndrome characteristics and how to encounter problems during anesthesia. This review assesses the limited number of studies available that address the therapeutic response to oral manifestations in RTT and the poor representativeness of the samples. Although it is understood that RTT is a rare disorder, prospective trials should use large cohorts of people with RTT in order to be either population-based or multi-center based in order to have greater representativeness. With respect to the overall quality of the published literatures, the authors feel that there is a fair amount of evidence available from the literature. A large majority of the articles we assessed were of decent quality in terms of quality of evidence, methodology, outcome and selection. Fourteen studies were assessed to be of Good Quality (7–9*) according to the Newcastle–Ottawa scale, seven studies were found to be of moderate quality (5–6*) and only one study was deemed to be of poor quality according to the scale. The authors believe that more high quality research will aid in understanding the oral implications of Rett syndrome better, which subsequently might help in delivering a more comprehensive assessment of oral findings from a larger sample size. The rarity of the disease can be one hurdle in achieving this goal. The evidence presented in the current systematic review, to the best of author’s knowledge, is the highest quality evidence present in the scientific literature pertaining to the review question.

## 5. Conclusions

There were 22 articles which met the research criteria. The small sample size and relatively small number of studies meant that the result cannot be described as completely representative, making it impossible to draw iron-clad conclusions. Rett syndrome patients face a lot of difficulties in their daily life due to restrictive mental and physical development. The willingness of Rett patients and their caregivers to follow oral hygiene instructions is critical due to oral hygiene complications associated with Rett syndrome. Dentists treating Rett syndrome patients should consider oral health education programs on special care dentistry. The most fundamental oral health manifestation of Rett syndrome is bruxism, which is difficult to treat. Understanding the characteristic oral manifestations, digital-oral habits and oro-facial motor functional issues associated with Rett Syndrome is critical for dental practitioners, in order to properly manage dental issues of Rett patients. Early diagnosis, oral health education and prompt treatment remains the key for treatment of oral complications associated with Rett syndrome.

## Figures and Tables

**Figure 1 ijerph-18-01162-f001:**
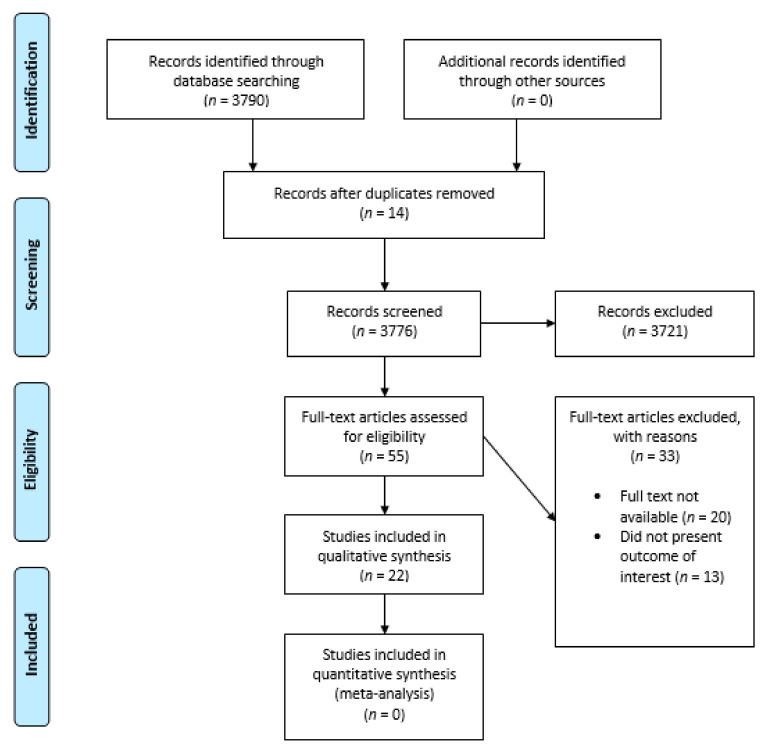
Prisma Flow Diagram.

**Figure 2 ijerph-18-01162-f002:**
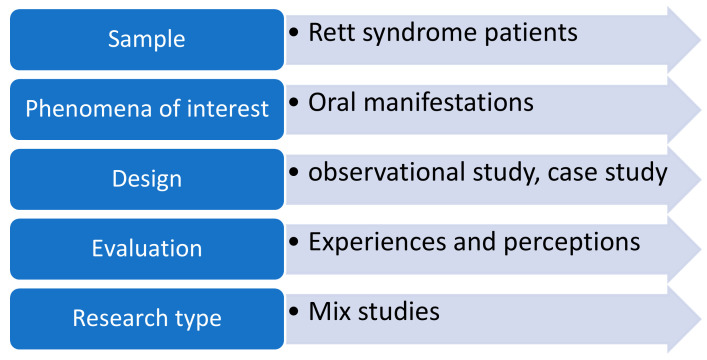
Spider Strategy.

**Table 1 ijerph-18-01162-t001:** Keywords and Search Strings.

Rett Syndrome	3790
Rett Syndrome and Oral Health	17
Dental Health of Rett Syndrome patients	13
Rett syndrome and dentistry	29

**Table 2 ijerph-18-01162-t002:** Inclusion and Exclusion Criteria.

Inclusion Criteria	Exclusion Criteria
Article published after 1990	Articles published prior to 1990
English language only	Articles not in English language
Original studies, review articles, case reports, case series	Editorials, opinions, correspondences
Only articles published in peer reviewed and indexed journals	Non peer reviewed/non indexed journals
Data bases examined (PubMed, CINAHL, Scopus, Medline, embase)	Little or no focus on dental aspects

**Table 3 ijerph-18-01162-t003:** Newcastle–Ottawa scale quality assessment form for non-randomized studies included in the review.

	Selection					Comparability		Outcome			Overall
Study & Year	1	2	3	4	5	6	7	8	9	10	Score (*)
Peak, 1992	*	*		*		*		*	*	*	7
Ribeiro, 1997	*	*		*	*	*	*			*	7
Alpoz, 1999	*	*			*	*		*		*	6
Kohyama, 2001		*	*			*	*	*		*	6
Magalhães, 2002		*	*			*	*	*			5
Khalil, 2002	*		*		*	*	*	*		*	7
Friedlander, 2003	*	*	*			*	*	*		*	7
Lavas, 2006	*	*	*			*	*	*	*	*	8
Alfonso, 2007	*	*	*		*	*	*	*	*		8
Green, 2008	*		*			*		*		*	5
Manish, 2010	*	*			*	*		*		*	6
Ji-sung, 2011	*	*	*			*	*	*		*	7
Fuertes, 2011	*	*	*		*	*	*	*		*	8
Morgan, 2012	*	*			*	*	*			*	6
Fuertes, 2014	*	*	*		*	*	*	*		*	8
Aleksandra, 2014	*	*	*	*	*	*		*	*		8
Janas, 2015	*	*		*		*		*	*		6
Marvin, 2015	*				*	*		*			4
Omer, 2016	*	*	*			*	*	*		*	7
Mezzedimi, 2017	*	*	*			*	*	*	*		7
Lai YYL, 2018	*	*	*			*	*	*		*	7
Pia, 2019	*	*	*	*	*	*		*	*	*	9

**Quality of studies**—Poor (0–4*), Moderate (5–6*) Good (7–9*); * Category Scoring criteria Selection—Representativeness of exposed group (Maximum score: 5 stars) low selection bias (1 star)—acknowledged selection bias (1 star)—Highly selected group (0 stars)—No description (0 stars) Selection of non-exposed group (maximum 1 star)—Same source population as exposed group (1 star)—Drawn from different source (0 stars)—No description (0 stars) Ascertainment of exposure (maximum 1 star)—Secure record (e.g., pathological record) (1 star)—Structured interview (1 star)—Written self-report (0 stars)—No description (0 stars) Outcome demonstrably absent at baseline (maximum 1 star)—Yes (2 Star) No (0 star); **Comparability**—Maximum 2 stars—Comparability of cohorts, controlling for confounders (Maximum score: 2 stars)—Controls for key confounders (e.g., Gleason grade) (1 star)—Controls for related factors (1 star)—Cohorts incomparable, confounders uncontrolled (0 stars); **Outcome assessment** (Maximum score: 3 stars)—Large studies/panels—secured records or directly measured (1 star)—self reported information (0 star), single target/objective (0 star). Adjusted for missing data of follow up (1 star). No follow up or statement about missing data (0 Star). Clear specification of outcomes (Yes—1 star, No—0 stars).

**Table 4 ijerph-18-01162-t004:** Oxford Center of Evidence-based Medicine (Levels of Evidence).

Level	First Author, Date, Reference
4	Peak, 1992 [14]
4	Ribeiro, 1997 [15]
5	Alpoz, 1999 [16]
5	Kohyama, 2001 [17]
4	Magalhães, 2002 [18]
4	Khalil, 2002 [19]
5	Friedlander, 2003 [20]
2b	Lavas, 2006 [21]
5	Alfonso, 2007 [22]
5	Green, 2008 [23]
4	Manish, 2010 [24]
4	Ji-sung, 2011 [25]
3a	Fuertes, 2011 [26]
1a	Morgan, 2012 [27]
3b	Fuertes, 2014 [28]
5	Aleksandra, 2014 [29]
5	Janas, 2015 [30]
4	Marvin, 2015 [31]
4	Omer, 2016 [32]
4	Mezzedimi, 2017 [33]
4	Lai YYL, 2018 [34]
1b	Pia, 2019 [35]

**Table 5 ijerph-18-01162-t005:** Dental aspects of Rett Syndrome, characteristics of studies reviewed.

First Author, Date, Reference	Type of Article	Sample Size	Article Key Points	Barriers and Limitations
Peak, 1992 [14]	Case report	5 year old girl	Severe bruxism-gross attrition of canines and incisors, Bilateral masseteric hypertrophy, No caries Patient education, Relaxation techniques, Lower polythene occlusal splint, Sedation or GA recommended.	None
Ribeiro, 1997 [15]	Case series	17 patients with mean age of 7.33	Bruxism (*n* = 14/17) Non-physiological dental attrition Digit/hand sucking or biting, mouth breathing, drooling, tongue thrusting. Palatal shelving, anterior open bite Gingivitis 76% (*n* = 13/17) 2.7% tooth surfaces had dental caries.	None
Alpoz, 1999 [16]	Case Report	5 year old girl	A girl with daytime bruxism. Dental wear, with near complete loss of the dental crown, albeit without exposure to pulp. The patient was offered nitrous oxide sedation and alginate samples were obtained to create a flexible splint for the upper jaw, to be used only when the parents were up and watched, because the patient was not bruxed at night.	Single case study
Kohyama, 2001 [17]	Review article	Review	Disturbance in phasic chin muscle activity during rapid-eye-movement sleep (REMS); an elevation of phasic inhibition index (PII), without disturbing of tonic inhibition index (TII) due to functional impairment of the pontine tegmentum.	Reports on autistic tendency in SMEI (severe nocturnal enuresis, autism) patients were not found.
Magalhães, 2002 [18]	Case series	13 patients age 9 years.	Update on the oral (bruxism) and general aspect (stereotyped hand movements, scoliosis) of the disorder. Oral hygiene instruction Dental prophylaxis. Application of topical fluoride. Construction of bite-plane. Acupuncture	Do not have the measure of the efficacy of bruxism treatment in Rett syndrome patients.
Khalil, 2002 [19]	Case report	7 year old girl	The girl weighed 14 kg, was weak, grinding her teeth constantly, and had a poor ability to open her mouth and stretch her neck. Importance of BIS monitor in inducing and maintaining level of anesthesia in children sensitive to drugs	Did not measure the wear on the teeth so do not have the measure of the efficacy of bruxism treatment in Rett syndrome patients.
Friedlander, 2003 [20]	Review	Review	Dentists caring for individuals with Rett, fragile X must be familiar with the manifestations of these diseases. They must also be familiar with the medications (anticonvulsants, anti-hypertensives, etc.) used to treat the associated behaviors as they might cause adverse reactions.	None
Lavas, 2006 [21]	Original Research	125 females with mean age of 19.6 years.	This study provides data describing Rett syndrome female difficulties from parents/caregivers aspect regarding eating, communication, functional and oral and dental problems.	The information from the study has been compromised to a small extent by the variation in the number of answers for different questions.
Alfonso, 2007 [22]	Original research	108 patients with mean age 31 years.	Protocol was planned and implemented coordinating with health care levels and workers, it provides necessary dental treatment to a large number of disabled people, who would not have received it otherwise.	It remains to be studied, the success or failure of treatments. A patient satisfaction and a cost analysis is still to be done.
Green, 2008 [23]	Review article	Review	Early interventions make the symptoms of the disorder less obvious. Management strategies for pediatric ASD population (Rett syndrome) describing behavioral and drug therapy.	Communication problem, dentists should have patience understanding of this neurological condition. Educating parents and patients in preventive care is imperative.
Manish, 2010 [24]	Case report	8 year old girl	Four stages of Rett’s syndrome have been defined to help characterize the disorder and improve its recognition and diagnosis. Kids with Rett’s syndrome frequently display autistic-like symptoms at an early stage. Certain signs may include jumping on the knees, sleeping problems, a wide-spread gait, teeth grinding and chewing difficulty, slow growth, hallucinations, cognitive disabilities, and breathing difficulties when waking up, such as hyperventilation, apnea, and air swallowing.	Limited case study
Ji-sung, 2011 [25]	Case report	19 year old girl	Many difficulties and issues for anesthetic management of RS patients, such as scoliosis and muscular tonicity and breathing abnormalities, ranging from centrally mediated hyperventilation to apnea, should be considered. Ante grade fiber optic-guided oro-tracheal intubation was possible to perform by an experienced anesthesiologist in patient with limited mouth opening.	Management could not be carried out in an effective and comprehensive manner.
Fuertes, 2011 [26]	Review article	35 cases of RS patients from 1985 to 2007	Drug-related oral manifestations include xerostomia, stomatitis, glossitis, erythema multiform, sialorrhea, gingivitis, dysphagia etc. Oral findings more prevalent in RS are bruxism, open bite, a high arched palate and gingivitis.	Patient communication capacity. Controversy in bruxism treatment, Sample being too small.
Morgan, 2012 [27]	Review article	Review	This review examined the effectiveness of interventions for oropharyngeal dysphagia in children with neurological impairment. The three studies included in the review examined oral sensorimotor treatments and lip strengthening interventions.	Insufficient randomized trials to evaluate the effectiveness of interventions for oropharyngeal dysphagia.
Fuertes, 2014 [28]	Case control study	RS Patients = 41 Mean age [13.37 (±) 3.19 years], Control match group = 82	Observational case-control study, followed the protocol of WHO for conducting health surveys. Data recorded by questionnaire and oral examination used to document caries indicators, CPI (Community Periodontal Index) and oral manifestations. Patients with RS: caries score was lower than the control group, they had increased frequency of periodontal problems, dental wear, drooling, high arched palate and anterior open bite. Oral habits of RS patients: diurnal bruxism, followed by stereotyped tongue movements and oral breathing.	Limited case study.
Aleksandra, 2014 [29]	Original research	37 RS female patients of 2–31 years and 34 Gender- matched Control group of 2–30 years	RTT participants have more difficulty reading emotional expressions and these problems are linked to atypicalities in scanning.	Single problem for each pairing of familiar/novel expressions and secondly, use of a single familiarization time for each problem.
Janas, 2015 [30]	Case report	18 year old girl	RS patients shall be sent to a medical center where appropriate medical and testing services and qualified personnel are available. Intra oral examination: teeth grinding—bruxism, difficulties in chewing and swallowing, neglect of oral cavity hygiene, oral inflammation with hypertrophic gingiva also many cavities in teeth. Extra oral examination showed hand wringing and washing movements, sudden lurching of the head towards the shoulder.	No specific recommendations on behavior guidance for the dental examination of RS patients.
Marvin, 2015 [31]	Review article	Review	Dentists may be the first healthcare providers to recognize that a 1- or 2-year-old child has some type of extraordinary pervasive behavioral disorder.	Treatment modalities are wide and tend to be of much error and no cure is present as these pervasive disorders present various entities. No reliable empirical biological tests (e.g., blood tests or brain scans) for ASD are reliable.
Omer, 2016 [32]	Case report	4.5 years old girl	Regional anesthesia should be preferred for suitable operations in patients with Rett syndrome since it reduces opioids, anesthetic requirements and postoperative respiratory depression. Patients with RS had a lower anesthetic dose requirement compared to the control group and suffered prolonged apnea at the postoperative period.	Utilization of BIS monitored not done to follow anesthetic depth and to titrate the effect of anesthetic agent.
Mezzedimi, 2017 [33]	Case study	61 female patients with mean age of 13.6 years.	Oral apraxia, dyskinetic tongue movements, prolonged oral stage, and poor bolus formation were the most common findings in all patients and progressive dysphagia was noted by caregivers. Useful instructions and suggestions for preventing dysphagia that facilitates in eating.	Single case reported.
Lai YYL, 2018 [34]	Case study	242 females.	Dental problems reported in severe genotype RS patients included gingival bleeding; and dental trauma from falls, bruxism or malocclusion. The incidence of restoration and extraction of teeth decreased with higher levels of income although extractions were more common than restorations.	Limited patient collaboration.
Pia, 2019 [35]	RCT	5 female patients.	Treatment with botulinum toxin (BTX) for hyper salivation is effective in reducing saliva production that may help in oral motor functions such as eating and bruxism.	Missing data in income level analysis, the allocated age for the incidence calculations had varied implications for analysis and comparison of the data and finally unclear formal diagnosis of GORD were made.

## Data Availability

Not applicable.

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
