# Peer review of "Oral Manifestations of Rett Syndrome—A Systematic Review"

_ijerph, 2021, doi:10.3390/ijerph18031162_

Round 1

Reviewer 1 Report

I would like to congratulate the authors for this paper, it is a great systematic review. I suggest double check the text formatting because there are some minor issues to fix. 

As a systematic review in my opinion it looks good, but perhaps it could be enriched by including more recent articles and improving the discussion section.

As a suggestion, the authors could develop a list or a table correlating dental issues and suggested treatments and also could explore how to perform the diagnose of this syndrome.

If the authors have at least one case report, it may be interesting to include it as well.

Reviewer 2 Report

Dear Authors 

the paper is interesting and well done.

The systematic review is well conducted. I observed some typing mistakes (in some rows Rett is not written in capital letters).

Make minor revisions to make the paper improved.

Reviewer 3 Report

I had the opportunity of revising the present systematic review about the oral manifestations of Rett syndrome, the article is quite well written, the literature in limited but for the same reason the review could be useful and original in describing the knowledge on the topic and encouraging further research.

Anyway, I list some changes to the manuscript that should be performed to consider the acceptance:

  • Have the authors followed the PICOS questions in describing the aim of the study? Please check if it is possible to improve the aim accordingly
  • The sentence at the end of the aim “…and help dental clinicians in understanding oral characteristics of Rett in order to formulate best practice treatments” seems a consideration that could be added to the discussion but is not strictly adherent with the scientific aim of the review. Please separate it from the actual aim or better move it to the discussion section.
  • In the methods section several sentences or information are repeated multiple times, please check it and revise.
  • Why “keyword should match keywords in the title or the abstract of the studies selected for our review” was an inclusion criterion? An article could be inherent the aim of the review even if the exact keywords are not in the title or abstract (for example synonyms could be used or the article could be related to a wider group of pathologies that includes the rett syndrome). This was also repeated multiple times in the manuscript.
  • How is possible that performing a research on more databases only 14 duplicates were found? (figure 1)
  • Why a separate paragraph title was used for exclusion criteria and not for inclusion criteria? I suggest to merge the paragraph with the previous one or to do a separate paragraph for both inclusion and exclusion criteria.
  • Please explain what the numbers under the headings “selection” “comparability” etc. means according to the Newcastle-ottawa scale adding a legend for table 3.
  • In the results, discussion and conclusion sections, the quality assessment of the level of evidence of each publication and of the whole production is completely absent. It is extremely important in a systematic review give results about the evidence level and appropriately comment it.
  • A part of the discussion section is related with the communication problem with patients with Rett syndrome. Please explain how this is related with dentistry or delete it.
  • The authors said that “Acid reflux has been generally described in rett patients (38) and may lead to dental erosion, but to date, no studies have directly linked non-physiological tooth wear to erosion due to gastroesophageal reflux.”, this is correct, but a few studies observed a higher risk of severe dental wear in subjects with association of bruxism and gastroesophageal reflux. Please check it and discuss it. Associations among Bruxism, Gastroesophageal Reflux Disease, and ToothWear. J Clin Med. 2018 Nov 6;7(11):417. doi: 10.3390/jcm7110417.
  • The last paragraph named “Why this paper is important to pediatric dentists?” seems to be related with a previous submission to a pedodontic journal and not appropriate to IJERPH, furthermore, the review has not the aim to analyze oral manifestations in children, so it should be removed.

Best regards

Round 2

Reviewer 3 Report

The authors clearly did an effort in improving the quality of the manuscript.

Some minor changes still needs to be performed before publication.

  • Regarding the Newcastle-Ottawa scale, the authors added the interpretation of the overall quality score, but the meaning of each point for achieving this score is still not explained. Please add what each star is referred to (for each number in the second line of the table).
  • In the discussion section the overall quality level of the published literature is still not discussed, you should discuss if the level is sufficient for giving significant information and what still lacks in term of quality and data that could be interesting for the reader.
